# A Technology-Driven, Healthcare-Based Intervention to Improve Family Beverage Choices: Results from a Pilot Randomized Trial in the United States

**DOI:** 10.3390/nu15092141

**Published:** 2023-04-29

**Authors:** Kristina H. Lewis, Fang-Chi Hsu, Jason P. Block, Joseph A. Skelton, Marlene B. Schwartz, James Krieger, Leah Rose Hindel, Beatriz Ospino Sanchez, Jamie Zoellner

**Affiliations:** 1Department of Epidemiology and Prevention, Division of Public Health Sciences, Wake Forest University School of Medicine, Winston-Salem, NC 27157, USA; 2Department of Biostatistics and Data Science, Division of Public Health Sciences, Wake Forest University School of Medicine, Winston-Salem, NC 27157, USA; 3Department of Population Medicine, Harvard Pilgrim Health Care Institute, Harvard Medical School, Boston, MA 02215, USA; 4Department of Pediatrics, Wake Forest University School of Medicine, Winston-Salem, NC 27157, USA; 5Rudd Center for Food Policy and Health, University of Connecticut, Hartford, CT 06103, USA; 6Department of Health Systems and Population Health, University of Washington School of Public Health, Seattle, WA 98195, USA; 7Healthy Food America, Seattle, WA 98122, USA; 8Department of Public Health Sciences, University of Virginia School of Medicine, Charlottesville, VA 22903, USA

**Keywords:** sugar-sweetened beverages, fruit juice, healthcare, intervention, family, randomized controlled trial, United States

## Abstract

**Background:** Healthcare-based interventions to address sugary beverage intake could achieve broad reach, but intensive in-person interventions are unsustainable in clinical settings. Technology-based interventions may provide an alternative, scalable approach. **Methods:** Within an academic health system in the United States that already performs electronic health record-based sugary drink screening, we conducted a pilot randomized trial of a technology-driven family beverage choice intervention. The goal of the intervention was to reduce sugar-sweetened beverage (SSB) and fruit juice (FJ) consumption in 60 parent–child dyads, in which children were 1–8 years old. The pediatrician-initiated intervention consisted of a water promotion toolkit, a video, a mobile phone application, and 14 interactive voice-response phone calls to parents over 6 months. The study was conducted between June 2021 and May 2022. The aim of the pilot study was to assess the potential feasibility and efficacy of the newly developed intervention. **Results:** Intervention fidelity was excellent, and acceptability was high for all intervention components. Children in both the intervention and the control groups substantially decreased their consumption of SSB and FJ over follow-up (mean combined baseline 2.5 servings/day vs. 1.4/day at 6 months) and increased water consumption, but constrained linear mixed-effects models showed no differences between groups on these measures. Compared to parents in the control group, intervention parents had larger decreases in SSB intake at 3 months (−0.80 (95% CI: −1.54, −0.06, *p* = 0.03) servings daily), but these differences were not sustained at 6 months. **Conclusion:** These findings suggest that, though practical to implement in a clinical care setting and acceptable to a diverse participant group, our multicomponent intervention may not be universally necessary to achieve meaningful behavior changes around family beverage choice. A lower-intensity intervention, such as EHR-based clinical screening alone, might be a less resource-intense way for health systems to achieve similar behavioral outcomes. Future studies might therefore explore whether, instead of applying a full intervention to all families whose children overconsume SSB or FJ, a stepped approach, starting with clinical screening and brief counseling, could be a better use of health system resources.

## 1. Introduction

Excessive consumption of sugar-sweetened beverages (SSBs) causes weight gain [1,2,3,4,5,6,7,8], dental caries [9,10,11], and other adverse health outcomes [12,13,14,15,16,17,18,19,20]. Despite these risks, SSBs continue to be over-consumed, contributing one-third of the added sugars consumed by American children [21,22] and negatively impacting diet quality [23]. One hundred percent fruit juice (FJ), though not a SSB, can also have negative health impacts—young children who overconsume FJ also drink more SSBs and have greater increases in weight during later childhood [24,25]. Facilitating healthy beverage choices in preschool and elementary school-aged children is therefore a major public health priority. This age range is particularly of interest, because young children innately prefer “sweet” over other flavors and are forming key dietary preferences that can be negatively impacted by excessive exposure to SSB [26]. This age range also represents a period of significant increase in the prevalence and volume of SSB and FJ consumption [27,28]. To assist parents and caregivers with making healthy choices for their families, nutritional guidelines recommend that young children avoid all SSBs, and that FJ are limited to one serving per day for children aged 1 year and older [29,30]. Guidelines for pediatric providers complement this advice, emphasizing the importance of routine screening and counseling about SSBs and FJ as part of a comprehensive approach to child obesity prevention [31,32,33].

With such an emphasis placed on child SSB and FJ screening and consumption in pediatric clinics, effective healthcare-based interventions to address child beverage intake could have broad reach and impact on population health [34]. Despite this potential, evidence for practical interventions implemented in clinical settings is lacking. Most prior beverage intervention studies have taken place in schools, communities, and home settings [35,36]. Successful behavioral interventions for beverage choice have been found to require high-intensity contact (10 h on average) and to engage parents as well as children [37]. Faced with a number of competing priorities, most clinical settings cannot realistically implement and sustain these types of high-intensity interventions targeting the whole family. Technology-based interventions may provide a feasible alternative, because they can be implemented without placing unrealistic time burdens on clinical staff or families [38,39,40,41]. However, it is not known whether a “lower-touch” technology-based intervention can be readily integrated into clinical care settings to improve child beverage intake. 

To begin to address this gap, we developed a novel healthcare-based intervention for family beverage choice. The goal of the present study was to test the feasibility and potential efficacy of our new intervention. We hypothesized that children in families provided with the intervention would have larger decreases in combined SSB and FJ consumption and complementary increases in water consumption over a 6-month period compared to those who received usual clinical care. Given the important role of parenting practices and role modeling for child beverage consumption outcomes, we included changes in parent beverage consumption, knowledge, attitudes, and beliefs, as well as intervention fidelity and acceptability, as secondary outcomes. 

## 2. Materials and Methods

### 2.1. Study Design

Between June 2021 and May 2022, we conducted a pilot parallel group, randomized controlled trial of a newly developed beverage choice intervention among 60 families whose children are primary care pediatric patients in an academic healthcare system. 

### 2.2. Setting

Participants were recruited from primary care pediatric and family medicine practices within the Atrium Health Wake Forest Baptist (AHWFB) system in the Piedmont region of North Carolina. This academic learning health system has an ongoing EHR-based program to screen all pediatric patients age 6 months through 17 years for SSB and FJ consumption [42]. The automated EHR screener is built as a best practice advisory for clinical support staff. The advisory prompts staff to ask about the frequency of SSB and FJ consumption while rooming pediatric patients, and is set to recur every 6 months. Responses are entered into the child’s EHR in an easily extractable data field for use in clinical care, research, and population health surveillance. For children whose EHR-based screening indicates higher than recommended levels of consumption, information about sugary drinks and healthy alternatives (e.g., water, small amounts of 100% fruit juice) is automatically added to the after-visit summary that is given to parents by the clinic team.

### 2.3. Study Population, Recruitment and Randomization, Reimbursement

Using EHR-derived SSB and FJ screening data, we identified potentially eligible 1–8-year-old children who were documented to be consuming 2 or more SSB and/or FJ servings per day. This age range was selected to include an age group with relatively uniform SSB and FJ intake recommendations; for this reason, we did not include adolescents. Parents of such children with a scheduled pediatric primary care visit within the next month were called by research staff to assess their interest and eligibility for study participation. Additional requirements for inclusion were having an English-speaking parent/caregiver with regular access to an internet-enabled device (e.g., smartphone, tablet, personal computer), and that child baseline total sugary drink (SSB + FJ) intake was confirmed by research staff to be ≥2 servings per day. We excluded families whose child had a medical condition that might render our dietary advice unsafe or inappropriate, as well as those who had been seen in a pediatric weight management program in the prior year (Figure 1).

All recruitment activities were performed by telephone, including obtaining informed consent for trial participation. One parent/caregiver provided informed consent per family. Per institutional policies, children aged 7 years and older also provided consent to participate. After informed consent was obtained, parents completed a baseline data collection interview by phone and were computer randomized in a 1:1 ratio to either the intervention or control group. REDCap (Research Electronic Data Capture) [43], a secure, web-based data management system, was used to build and manage a randomization schema. The randomization list was created by a permuted block algorithm with random block lengths. Randomization was stratified by parent race/ethnicity to ensure balance between groups. Parents (one per family) were provided with a $25 gift card upon completion of baseline data collection and 3-month data collection and a $50 gift card upon completion of 6-month data collection ($100 total per family). All post-randomization assessments were conducted by a trained research assistant blinded to participant study group allocation. Data were also captured and stored using the REDCap system. The protocol and procedures for the trial were reviewed and approved by the Wake Forest University Health Sciences Institutional Review Board (IRB# 00062659), and the study was registered at ClinicalTrials.gov, accessed on 14 May 2021 (NCT04886817).

### 2.4. Intervention Rationale and Development

The Revised Family Ecological Model by Davison and colleagues provided the behavioral model for our intervention [44]. The model postulates that to address child beverage consumption, one must also change parental knowledge, beliefs, and possibly behaviors and attempt to address upstream barriers or stressors that might inhibit parental knowledge and belief changes. To identify relevant barriers or stressors that could impede behavior change around drink choices and to refine intervention content and delivery processes, we conducted semi-structured interviews with a separate sample of parents in the year before the pilot study (data and qualitative analysis reported in a separate manuscript). 

Using data from these parental interviews, combined with published evidence from prior successful similar health behavior interventions, we developed an intervention consisting of four components (See Appendix A for detailed descriptions and images). The first component was a water promotion toolkit to increase the appeal of water drinking [45]. The toolkit was provided by a child’s pediatrician in a cloth tote bag bearing the study logo and included colorful reusable dishwasher-safe water bottles, stickers, and a children’s book about drinking water [46]. The second intervention component was a 5-minute educational video about family beverage choice, which leveraged narrative persuasion (or storytelling) and a reality TV-style approach, based on successful health communication campaigns to change sugary drink consumption or other health behaviors [47,48]. Given the age of the participants (parents of young children, pre-school and elementary-school-aged children) and the ubiquity of smartphones in this population, the third component was a mobile-phone application (app) called “Ready, Set, Gulp!”, which assisted parents and children with goal setting, tracking of SSB and FJ, and water intake [49,50]. It included activities such as a trivia function and a “Sugar Translator”, so that users could explore the sugar content in drinks. The final intervention component was a series of 14 computerized interactive voice-response (IVR) phone calls made to parents over a 6-month period to help support goal setting and address potential barriers or facilitators of behavior change (e.g., concerns about tap water safety, how to engage other family members in change) [51].

Research staff notified clinical providers and practice managers via an EHR in-basket message when an intervention family was scheduled for a clinic visit. The message included standardized instructions to providers to give out the toolkits during the child’s scheduled medical visit and to conduct brief counseling with parents around using the intervention. A welcome pamphlet within the toolkit further instructed parents on how to access the remaining intervention components (video, app, and phone calls) by scanning a QR code with their smartphones.

### 2.5. Control Group

Families randomized to the control group received usual care. As noted previously, within the AHWFB health system, usual care includes a prompt for clinical team members to conduct EHR-based screening for SSB and FJ consumption at pediatric visits. To improve retention of control group participants in this study, research team members also sent monthly text messages to parents over the 6 months of follow-up to confirm contact information and ongoing engagement in the study. After completing the final data collection, control families were mailed a water promotion toolkit.

### 2.6. Outcome Measures

The primary outcome measure for this pilot was change from baseline in a child’s combined consumption of SSB + FJ, measured in terms of frequency per day, at 3 and 6 months after randomization. Beverage intake was estimated using the BevQ15 instrument, completed by the parent on behalf of their child [52]. This instrument has been validated and used in diverse populations [52,53] and can be administered to adults for proxy report of a child’s consumption. As secondary outcomes, we assessed changes in total daily volume (ounces) of SSB + FJ, as well as changes in the child’s water intake (frequency and volume), also using the BevQ15 instrument. We also measured parental beverage consumption with the BevQ15. 

Additional secondary outcomes included parent knowledge, attitudes, and beliefs around family beverage choices. We captured this information at baseline and 6 months using a survey that included items that assessed parental awareness of guidelines around the frequency of SSBs and FJ consumption, as well as items that assessed parental ability to correctly categorize beverage examples. The survey also assessed parental perceptions of various labeling and health claims found on drink packaging [54]. It included items from the Summer Styles survey [55] focused on awareness of potential health consequences of added sugar consumption and personal beliefs around what children should drink. Scoring all knowledge items as correct or incorrect yielded a total possible score range of 0–22 points, with subscores for awareness of recommended levels of FJ and SSB consumption by child age (0–6 points), correct identification of beverage category (0–10 points), and correct identification of health conditions related to sugar consumption (0–6 points). Parental rating of the perceived healthfulness of different beverage types was scored on a 1 to 10 scale (least healthy to most healthy). Similarly, parental rating of the importance of different health claims on drink packaging was rated from 1 to 10 (least important to most important). 

### 2.7. Fidelity and Acceptability Measures among Intervention Participants

We tracked several measures of intervention fidelity, including whether toolkits were received by each family, whether parents watched the video, downloaded the mobile phone app, and completed each of the 14 IVR calls. 

Intervention group parents were invited to complete an exit survey after finishing their 6-month data collection measures. This survey was conducted by an unblinded team member to allow for the collection of information on intervention acceptability, effectiveness, and use patterns for each intervention component. For the mobile phone app, parents also completed the 10-item Systems Usability Scale (SUS) measure, which has a total score range of 0–100, with higher scores reflecting a more user-friendly interface (e.g., >72.6 is considered good or above average, and >85.5 is excellent) [56]. 

### 2.8. Other Measures

At baseline, household structure/size, self-reported race/ethnicity for the parent and child, and parental education level were assessed by interviewer-administered surveys. The Single-Item Literacy Screener [57] and a modified version of the Mobile Device Proficiency Questionnaire [58] were used to understand how well participants could engage with our technology-based intervention. EHR-derived measures included child age (years) at baseline and sex. 

### 2.9. Analysis

Measured baseline characteristics for all participants were summarized by arms. To ensure the conditional normality assumption was satisfied, the distributions of outcome measures were examined. Appropriate transformations were performed if needed. Between-group changes in all outcomes were compared using constrained linear mixed-effects models, accounting for visit (baseline, 3 months, and 6 months) and intervention group by visit interaction as independent variables. This model uses all available longitudinal data and imposes a constraint of a common baseline mean between groups as a result of randomization [59]. Changes between follow-up and baseline measures were compared for each arm. Beverage consumption outcomes were captured and modeled at baseline and at 3 and 6 months, and knowledge, attitude, and belief outcomes and changes in DSQ components were captured and modeled only at baseline and 6 months. Analysis was performed in an intent-to-treat fashion (i.e., participants were analyzed based upon groups they were randomized to, regardless of subsequent level of engagement with intervention).

We explored effect modification by race/ethnic group by adding race/ethnic group and interaction between race/ethnic and intervention groups in the constrained linear mixed-effects models for main beverage outcomes. Any statistically significant interactions would indicate that the intervention effect is different by race/ethnic group. Further, given that our intervention period spanned multiple seasons, and in light of the known association between beverage intake and season (more SSB and water are consumed during warm weather months), we ran additional models adjusting for season (winter or spring vs. summer or fall) of each measure. Multiple comparisons were not corrected due to the fact that this was a pilot study. A *p*-value less than 0.05 was considered statistically significant.

## 3. Results

### 3.1. Participant Characteristics and Retention

In total, 413 families were reached by phone to attempt recruitment. Of these, 78 (19%) were found to be ineligible. Of the remaining 335 families reached, 273 (81%) declined participation, 2 (0.6%) completed consent but did not complete the baseline visit (so were not randomized), and 60 (18%) were successfully recruited and randomized (Figure 1). Among those who declined participation, commonly cited reasons included not being interested and being too busy to participate in a 6-month-long study.

Among the 60 families enrolled in the study, the mean (sd) age for parents was 34.1 (9.3) years and 4.2 (2.1) years for children (Table 1). A total of 14 (23%) children were under the age of 2, 22 (37%) children were between 2 and less than 5 years old, and 24 (40%) children were greater than or equal to 5 years old. A total of 43% of parents self-reported as non-Hispanic Black, 30% as non-Hispanic white, and 27% as Hispanic. The vast majority (88%) of parents involved in the study were mothers, and 66% of parents reported having attained some education beyond high school. Baseline mobile device proficiency was high, with a mean (sd) MDPQ score of 38 (3.2) out of a maximum possible 40 points. Over the 6-month study period, six families in the intervention group and two in the control group either withdrew or were lost to follow-up, such that at the final data collection visit, intervention group retention was 80%, and control group retention was 93%. 

### 3.2. Intervention Fidelity and Acceptability

All 30 families in the intervention group received their water promotion toolkits, although several received them by mail, because the clinics forgot to give them out during the pediatrician visit or due to the nature of the visit as a telehealth visit due to COVID infection. A total of 29 out of 30 (97%) parents watched the video, and 22 of 30 (73%) downloaded and installed the mobile phone application on their personal device. All 30 intervention parents completed at least one IVR phone call, with 26 of 30 (87%) completing at least half of the calls in the 6-month series. Call completion rates declined over the intervention period, with 97% completing call #1, 73% completing call #8, and 53% completing call #14 (Appendix A). Overall, across 420 attempted calls, 299 (71%) were completed. 

Among the 24 intervention group parents who completed the exit interview, all intervention components were rated favorably. The toolkit earned the best average score for overall parent-rated usefulness for changing family beverage choices, followed by the video, IVR calls, and app (Table 2 and Appendix A). The mobile phone app had a mean (sd) SUS score of 74 (14), indicating good but not excellent usability. 

### 3.3. Child Beverage Consumption Outcomes

Children in both the intervention and control groups decreased their reported frequency and total volume of SSB and FJ consumed over follow-up (Figure 2). For example, at baseline, mean (sd) daily frequency SSB and FJ intake was 2.4 (1.6) among intervention children and 2.6 (1.8) among controls. At month 6 of follow-up, it had declined to 1.3 (1.4) among intervention children and 1.5 (1.2) among controls. Conversely, reported frequency and total volume of water consumed increased somewhat over follow-up (Figure 2). At baseline, mean (sd) daily volume of water was 16.6 (11.4) ounces among intervention children and 15.5 (13.0) ounces among controls, and at the 6-month follow-up, it had increased to 19.7 (12.1) among intervention and 16.7 (12.0) among controls. There were no statistically significant between-group differences for changes in servings or volume of SSBs, FJ, or water for children at the 3- or 6-month data collection points (Table 3).

### 3.4. Parental Beverage Consumption Outcomes

Parents in both groups also decreased their reported frequency and total volume of SSBs and FJ consumed over follow-up and increased their reported frequency and volume of water intake (Figure 3). At the 3-month follow-up, intervention group parents reported a significantly greater decrease in the frequency of combined SSB/FJ intake (−0.80 fewer servings per day, *p* = 0.035) and SSB intake (−0.78 fewer servings per day, *p* 0.017) versus control parents. These between-group differences were no longer present at the 6-month follow-up. There were no statistically significant between-group differences for change in the volume of SSBs, FJ, or servings or volume of water for parents at the 3- or 6-month data collection points (Table 3).

### 3.5. Parental Knowledge, Attitude, and Belief Outcomes

Baseline parental knowledge scores were highest for awareness of health conditions related to sugar consumption and lowest for familiarity with child beverage intake guidelines. Overall knowledge scores changed very little in either group from baseline to the 6-month follow-up (Table 4). For intervention parents, the mean (sd) knowledge score at baseline was 14.9 (3.9) points (of a maximum of 22), increasing to 15.6 (3.3) at 6 months. For control parents, the baseline knowledge score was 14.0 (3.5) points, increasing to 14.5 (3.1) at 6 months. There were no differences in the change in overall knowledge scores or domain knowledge scores between groups. Similarly, there were few differences in the changes for ratings of the perceived healthfulness of different drink types or the importance of different health claims. Exceptions were that at the 6-month follow-up, intervention parents rated “Snapple Iced Tea” and “Capri Sun” as less healthy drink options compared to control parents (−1.2 points lower, *p* 0.012 for Snapple, 1.3 points lower, *p* 0.015 for Capri Sun), and they placed greater importance on “calories” as a feature used to select beverages for their child (2.4 points higher, *p* 0.006).

### 3.6. Sensitivity Analyses

There was no interaction between child race/ethnicity and intervention effect for the examined outcomes (all *p*-values > 0.22). Specifically, the *p*-value for interaction between race/ethnicity and intervention on child SSB/FJ frequency was 0.53. Similarly, adjusting for the seasonality of data collection did not change the findings—both model outputs that were significant (i.e., 3-month parental change in SSB/FJ and SSB frequency) and those that were non-significant in our primary models remained so, with very little change in the magnitude or direction of parameter estimates.

## 4. Discussion

In this pilot randomized trial, a technology-driven, health system-based family beverage choice intervention was found to be acceptable and feasible. Adult and child participants in both the intervention group and the control group self-reported favorable changes in their beverage consumption patterns over the six-month study period, with no between-group differences for children. Parental SSB consumption in the intervention group did decline more than the control at 3 months, but this difference was not sustained at 6 months. The control condition consisted of monthly parental outreach from the research team and clinic-based, EHR-driven beverage screening and information provided in printed after-visit summaries to parents.

Our work builds on many prior studies that have examined strategies to address the overconsumption of sugary drinks and to promote water intake in families. A 2017 systematic review and meta-analysis by Vargas-Garcia and colleagues [37] reviewed 40 such studies. They found reductions in child sugary drink consumption for interventions conducted in schools or community settings, but not for those introduced in clinical settings. Overall, reductions in reviewed studies were clinically modest, averaging approximately 2oz less SSB consumed per day across successful interventions. In contrast, a 2018 review by Vercammen et al., which focused solely on interventions in early childhood, identified 11 studies associated with healthcare settings, 7 of which reported improvements in parental or child beverage consumption behaviors [34]. Many of the identified successful “healthcare” interventions actually relied heavily upon providers or interventionists outside of the healthcare system (e.g., community health workers [60], WIC staff [61], childcare providers [62]) or providers who may not be present in most US primary care settings (e.g., dietitian, psychologist [63], dentist [64]). In contrast, interventions that were delivered entirely by primary care medical providers were generally quite brief [65,66,67,68,69] and resulted in less behavior change [65,67,68]. The present study was designed to build upon this prior work by creating an intervention that could be sustainably translated into routine primary care pediatric practice, without the need for hiring additional staff or relying on interventionists outside of the clinic.

A major goal of this pilot study was to establish whether our technology-based beverage choice intervention was feasible in our clinical setting and acceptable to a diverse patient population. Thus, a strength of the study was that the participant population was predominantly Black or Hispanic and was representative of the clinical population in our health system who have previously been identified to be at greater risk of over-consuming SSB and FJ [42]. Supporting acceptability of the intervention, parents rated all intervention components favorably. Further, despite the relatively low-touch nature of the study, intervention fidelity was high. The mobile phone application, which we had envisioned as the core of our intervention to support behavior change, had the lowest level of uptake among intervention families, with just 73% of parents downloading it (compared to almost 90% who completed at least half of the IVR calls). Additionally, a plurality reported that their family’s use of the app waned over the intervention period. Although app feedback was positive overall, parents expressed slightly less enthusiasm for it than for other parts of the intervention. It is possible that further improvements to the app (e.g., more animation, better graphics) or more dedicated parental coaching on how to download and use it might enhance its fidelity and utility and potentially promote more durable behavior change. Additionally, in this small pilot, we were unable to explore whether differences in fidelity drove differences in behavior changes among intervention group participants.

Our top-line finding in this pilot was null. Although the direction of effect for behavior changes generally favored the intervention group and was consistent across multiple outcome measures, there were no statistically significant differences between intervention and control children for change in SSB, FJ, or water intake at the end of the 6-month follow-up. Both groups had clinically meaningful decreases in sugary drink intake [70,71] and increases in water intake. Parents exposed to the intervention did report larger decreases in sugary drink frequency at 3 months compared to those in the control arm. This is a positive and important initial finding, because changing parental behaviors, and thus facilitating the modeling of healthy behaviors, is a critical step for family-based beverage interventions [37]. It is possible that we observed an intervention effect among parents but not among children because children in this study were very young (mean age 4 years). As a result, parents were the ones more directly engaged in the intervention (e.g., IVR calls were made to parents, parents watched the video, and parents installed the app on their phones). Modifying the intervention to help parents better engage their young children in using the water promotion toolkit, reading the book, and using the child-friendly mobile phone app might improve reported child behavior changes. Related to this, a potential limitation of our pilot study is the wide range in ages and developmental stages of the children included. Future studies might examine whether narrowly focusing on school-aged children (i.e., omitting toddlers and preschoolers) could increase the reported impact of the intervention on child behavior.

An important and likely contributor to the finding of “no difference” between intervention and control children was that the control group also reported large decreases in child sugary drink consumption over follow-up. This finding could represent regression to the mean, or it may suggest that simply being in the study, with repeated outreach and beverage assessments with our study team to improve retention, may have sufficiently motivated control parents to change their child’s beverage intake. Additionally, our clinical context for this study was unique. As noted previously, our health system has implemented a routine point-of-care EHR-based sugary drink screening process [42]. This means that control group parents and children were likely being made aware of the importance of beverage choice by pediatric providers and could have received more pediatrician-based counseling about this issue compared to families in most primary care settings. A true no-contact control group in a different clinical setting might have reported less improvement over follow-up. Alternatively, the null finding could simply be due to the small sample size of our pilot, which was mostly geared toward feasibility and underpowered for finding small between-group differences. A post-hoc power calculation for our primary outcome (6-month change in child SSB/FJ frequency per day) suggested that we would have needed at least 84 participants (42 per group) to have the power to detect the observed between-group difference of 0.96 servings per day for children. The null finding could also reflect the inherent limitation of self-report dietary behaviors if there was a social desirability bias that led parents to systematically under-report child sugary drink intake during follow-up data collection.

The changes in behavior that we observed in this study (e.g., a decrease in daily child SSB/FJ consumption of 6–8 ounces), in both the intervention and control groups, occurred despite essentially no observed change in parental knowledge, attitudes, and beliefs about family beverage choice. At baseline, most enrolled parents were already quite knowledgeable about the health impacts of sugary drinks and believed them to be unhealthy options for their families. The intervention did not improve this knowledge, perhaps because there was not much improvement to make. This finding highlights an important point about the potential emphasis of future beverage choice interventions aimed at families. It suggests that many parents do not need a lot of help or training understanding the “why” of reducing their family’s sugary drink intake and increasing water intake. Instead, interventions focused on “when” and “how” to make these changes might be more productive. As such, future intervention adaptations could consider incorporating a more robust set of evidence-based behavioral change techniques [72,73,74]. Future studies might also explore whether, instead of applying a full multi-component intervention to everyone, using adaptive trial designs or a micro-randomization approach could be a better use of health system resources [75]. Under such a model, healthcare systems could begin with EHR-based SSB screening for everyone, as our control group received, and intensify intervention as needed, with additional components reserved for children flagged as persistent high consumers. Similarly, future studies may also apply a factorial experimental design or multiphase optimization strategy (MOST) to determine which combination of the multicomponent intervention strategies (e.g., video, app, and/or phone calls) may be most effective and cost effective for improving child beverage consumption [76].

One important final note about this pilot study was that it took place during the COVID-19 pandemic (recruitment during Spring and Summer of 2021). Despite this challenging clinical and societal context, we met recruitment and retention goals and had high intervention fidelity. Comparing our enrollment rate to prior studies is important, as this can be an important indicator of feasibility and external validity. However, participation rates are often underreported across intervention studies. For example, in a systematic review of 55 SSB interventions for children and adolescents, only (29%) reported this information; the mean participation rate was 66% [36]. Yet, most reviewed studies (71%) were school-based interventions, with only 7% of reviewed studies taking place in clinical settings, which limits comparison to our study. In the Vercammen review of SSB interventions, 11 studies were conducted in healthcare settings, yet only two were trials recruiting individual patients that provided recruitment rates [34]. In the study with a healthcare setting recruitment and intervention most similar to ours, about 20% of eligible families were enrolled [63], which is comparable to our enrollment rate of 18%. An important potential methodologic consideration for calculating the participation rate is that in studies recruiting with flyers or community advertisement, the true denominator of people “approached” for participation is unknown, and thus rates are calculated based on the likely small subset of individuals who contact the study team to initiate eligibility screening. Our total denominator for “people approached” is known, which makes our participation rate appear lower than these other types of studies whose denominator is not representative of all people who were exposed to the recruitment modality (e.g., read a flyer). Nonetheless, future efforts are needed to better understand potential barriers and opportunities optimizing recruitment rates in healthcare settings. We did note slightly higher loss to follow-up in the intervention group for this pilot. Given the overall high levels of acceptability reported for this technology-based intervention, dropout seems unlikely to be due to excessive participant burden; however, it will be important to explore this finding in the context of a larger study.

## 5. Conclusions

Overconsumption of sugary drinks—and added sugars more generally—remains an important health issue for young children, placing them at risk of numerous adverse health consequences. Our findings suggest that technology-based interventions to improve beverage choice can be feasibly implemented in busy health systems serving diverse patient populations, even in the midst of challenging circumstances such as those produced by a global pandemic. However, given that child sugary drink intake in our study improved regardless of study arm, our findings also suggest that a lower-intensity intervention, such as EHR-based clinical screening alone, might be an equally effective and less resource-intense way for health systems to achieve similar behavioral outcomes.

## Figures and Tables

**Figure 1 nutrients-15-02141-f001:**
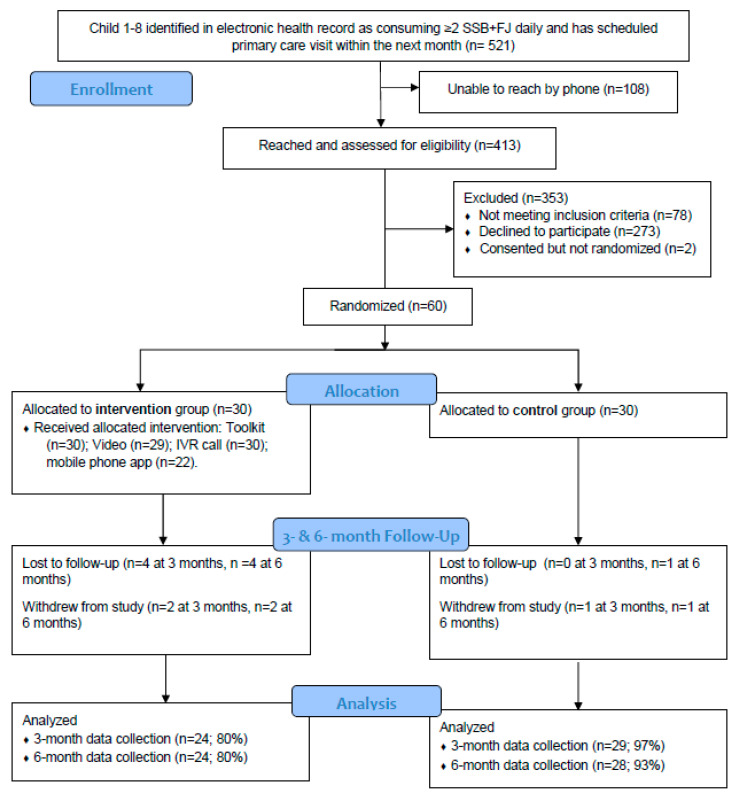
CONSORT flow diagram for enrollment.

**Figure 2 nutrients-15-02141-f002:**
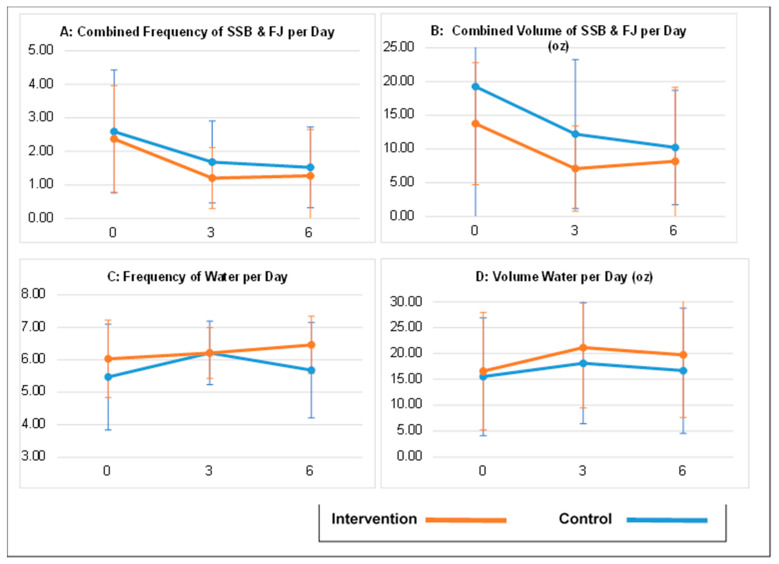
Child Beverage Consumption at Baseline, 3 and 6 months of Follow-up ^a^. a—For each outcome, figures present mean (+/−sd) by group at baseline (0 months), 3 and 6 months of follow-up, as calculated based on parental response to the Bev-Q-15 questionnaire. Panel **A** shows change over time for reported total times per day a child consumed a sugar-sweetened beverage or fruit juice. Panel **B** shows change over time for children’s total combined volume, in ounces of sugar-sweetened beverage and fruit juice consumed per day. Panel **C** shows change over time for reported total times per day a child consumed water. Panel **D** shows change over time for children’s total volume, in ounces, of water consumed per day. As noted in Table 3, there were no statistically-significant differences between groups for change from baseline for any outcome at either the 3 or 6 month mark.

**Figure 3 nutrients-15-02141-f003:**
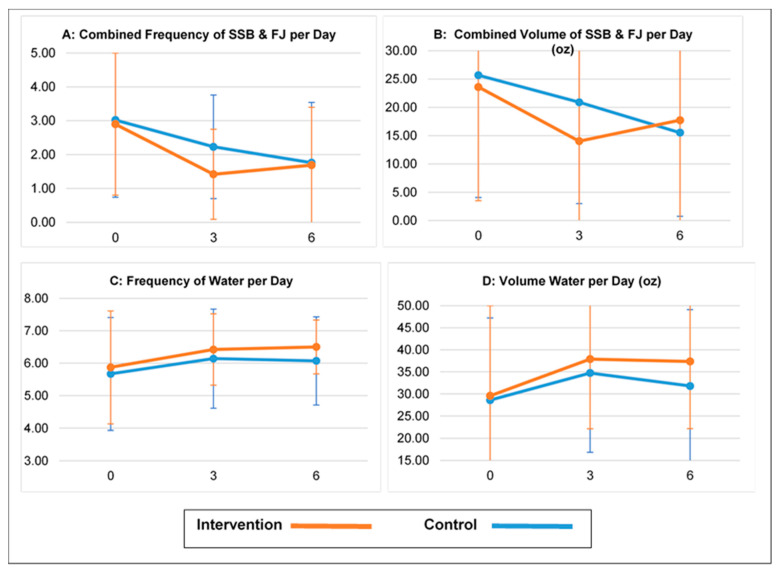
Parental Beverage Consumption at Baseline, 3 and 6 months of Follow-up ^a^. a—For each outcome, figures present mean (+/−sd) by group at baseline (0 months), 3 and 6 months of follow-up, as calculated based on parental response to the Bev-Q-15 questionnaire. Panel **A** shows change over time for reported total times per day a parent consumed a sugar-sweetened beverage or fruit juice. Panel **B** shows change over time for parental total combined volume, in ounces of sugar-sweetened beverage and fruit juice consumed per day. Panel **C** shows change over time for reported total times per day a parent consumed water. Panel **D** shows change over time for parental total volume, in ounces, of water consumed per day. As noted in Table 3, intervention parents had significantly (*p* < 0.05) larger decreases in combined sugar-sweetened beverages and fruit juice frequency at month 3 compared to control parents. There were no other between-group differences for change in parental beverage measures over follow-up.

**Table 1 nutrients-15-02141-t001:** Baseline characteristics of participating families in the “Ready, Set, Gulp!” pilot trial.

Characteristic	Control (n = 30)	Intervention (n = 30)	Overall (n = 60)
Parental age^a^ (mean (sd); years)	32.9 (9.0)	35.2 (9.6)	34.1 (9.3)
Child age ^b^ (mean (sd); years)	3.8 (2.2)	4.5 (2.1)	4.2 (2.1)
Race/ethnic group (n/%) ^a^ Non-Hispanic Black Non-Hispanic White Hispanic	13 (43%) 9 (30%) 8 (27%)	13 (43%) 9 (30%) 8 (27%)	26 (43%) 18 (30%) 16 (27%)
Marital status ^a^ Married Divorced/separated/other	13 (43%) 17 (57%)	10 (33%) 20 (67%)	23 (38%) 37 (62%)
Female Child (n (%)) ^a^	11 (37%)	16 (53%)	27 (45%)
Parental role is mother ^a^	28 (93%)	25 (83%)	53 (88%)
Parent education greater than highschool (n (%)) ^a^	19 (66%)	19 (66%)	38 (66%)
MDPQ score ^c^ (mean (sd))	37.9 (3.9)	38.1 (2.4)	38.0 (3.2)

a—Determined by parental self-report using pre-randomization telephone survey; b—based on date of birth in the electronic health record; c—mobile device proficiency score (as detailed in methods section), range of 0–40 points.

**Table 2 nutrients-15-02141-t002:** Overall intervention feedback from parents participating in exit survey.

Intervention Component	Mean (sd) Parental Rating: Overall Helpfulness for Decreasing Their Family’s Sugary Drink Intake ^a^	Mean (sd) Parental Rating: Overall Helpfulness for Increasing Their Family’s Water Intake ^a^
“Get in the Zero Zone!” video	3.0 (1.9)	3.0 (1.2)
Water promotion toolkit	3.5 (0.7)	3.6 (0.6)
“Ready, Set, Gulp!” mobile phone app	2.3 (1.2)	2.4 (1.2)
IVR call series	3.0 (1.2)	3.0 (1.1)

a—Based on telephone-based exit survey among 24 intervention group parents. Parents were asked to rate the overall helpfulness of each intervention component on a 1 to 5 scale, with 1 being the worst and 5 the best, for changing their family’s beverage choice behaviors.

**Table 3 nutrients-15-02141-t003:** Differences for change in beverage intake at 3 and 6 months of follow-up, comparing intervention to control participants.

Beverage Intake Metric ^a^	Difference at 3 Months ^b^ (95% CI; *p*-Value)	Difference at 6 Months ^b^ (95% CI; *p*-Value)
Children		
Combined SSB/FJ frequency ^c^	−0.42 (−1.01, 0.16; 0.15)	−0.27 (−0.97, 0.42; 0.43)
Combined SSB/FJ volume ^d^	−4.50 (−9.53, 0.54; 0.08)	−1.50 (−6.69, 3.69; 0.56)
Water frequency	0.22 (−0.17, 0.60; 0.27)	0.39 (−0.10, 0.88; 0.12)
Water volume	2.55 (−2.77, 7.86; 0.34)	2.06 (−4.05, 8.17; 0.50)
SSB frequency	−0.29 (−0.73, 0.14; 0.18)	−0.01 (−0.46, 0.44; 0.95)
SSB volume	−2.40 (−5.88, 1.08; 0.17)	0.21 (−3.67, 4.09; 0.91)
FJ frequency	−0.11 (−0.52, 0.30; 0.60)	−0.25 (−0.69, 0.19; 0.26)
FJ volume	−2.08 (−6.18, 2.03; 0.32)	−1.86 (−4.42, 0.70; 0.15)
Parents		
Combined SSB/FJ frequency	−0.80 (−1.53, −0.06; 0.03)	−0.27 (−1.05, 0.51; 0.49)
Combined SSB/FJ volume	−6.04 (−15.62, 3.54; 0.21)	0.89 (−6.96, 8.73; 0.82)
Water frequency	0.01 (−0.47, 0.49; 0.96)	0.19 (−0.16, 0.55; 0.27)
Water volume	1.87 (−6.16, 9.90; 0.64)	4.69 (−2.15, 11.53; 0.18)
SSB frequency	−0.78 (−1.42, −0.14; 0.02)	−0.26 (−0.91, 0.39; 0.42)
SSB volume	−4.96 (−13.45, 3.54; 0.25)	1.38 (−5.31, 8.06; 0.68)
FJ frequency	−0.01 (−0.27, 0.25; 0.95)	−0.02 (−0.27, 0.24; 0.89)
FJ volume	−1.36 (−5.10, 2.38; 0.47)	−0.51 (−3.06, 2.04; 0.69)

a—Based on responses to Bev-Q-15 questionnaire; b—comparing intervention to control group participants, in constrained linear mixed models accounting for visit and arm by visit interaction; c—frequency in number of times consumed per day; d—volume in ounces per day.

**Table 4 nutrients-15-02141-t004:** Knowledge, attitude, and beliefs of parental participants at baseline and 6 months of follow-up—descriptive results and comparison of change between groups with constrained linear models.

Survey Measure (Domain or Item) ^a^	Baseline Score/Value (Mean (sd))		6-Month Score/Value (Mean (sd))		Difference at 6 Months ^b^ (95% CI; *p*-Value)
	Control	Intervention	Control	Intervention	
Knowledge Items					
Identification of health conditions related to sugar intake (range 0–6 points)	4.7 (1.9)	4.8 (1.7)	5.2 (1.5)	5.4 (1.7)	0.17 (−0.71, 1.05; 0.70)
Classification of beverage examples (range 0–10 points)	7.1 (1.5)	7.7 (1.5)	7.2 (1.5)	7.8 (1.4)	0.29 (−0.47, 1.05; 0.45)
9–0.47,Identification of age-specific guidelines for child SSB and fruit juice intake (range 0–5 points)	2.2 (1.6)	2.4 (1.7)	2.1 (1.5)	2.5 (1.5)	0.18 (−0.50, 0.87; 0.59)
Total knowledge score (range 0–21 points)	14.0 (3.5)	14.9 (3.9)	14.5 (3.1)	15.6 (3.3)	0.51 (−1.02, 2.04; 0.51)
Attitude or Belief Items					
Rating of healthfulness for Capri Sun (1–9 scale, 1 = worst or least healthy, 9 = best or most healthy)	3.3 (2.5)	3.5 (2.2)	3.6 (2.4)	2.4 (1.7)	−1.18 (−2.13, −0.23; 0.02)
Rating of healthfulness for Snapple Iced Tea (1–9 scale, 1 = worst or least healthy, 9 = best or most healthy)	2.1 (1.9)	2.1 (1.9)	3.0 (2.2)	1.8 (1.1)	−1.16 (−2.06, −0.26; 0.01)
Level of agreement with statement: “Most children need to drink sports drinks like Gatorade or Powerade when they are active to keep them hydrated.” (1–5 scale, 1 = strongly disagree, 5 = strongly agree)	3.3 (1.5)	3.0 (1.6)	3.5 (1.3)	2.4 (1.6)	−0.96 (−1.65, −0.27; 0.01)
Level of agreement with statement: “Energy drinks such as Red Bull or Monster are a type of sports drink, such as Powerade or Gatorade.” (1–5 scale, 1 = strongly disagree, 5 = strongly agree)	1.6 (1.3)	1.6 (1.0)	1.3 (1.0)	2.1 (1.7)	0.78 (0.02, 1.54; 0.04)

a—For knowledge items, individual items are not shown, but rather total knowledge scores across several domains: identifying beverage types correctly (e.g., water, sugar-sweetened beverage, or 100% fruit juice), familiarity with child beverage intake guidelines, identification of health conditions associated with sugar consumption, and total knowledge correct. For attitude and belief items, these are presented as individual items. The table includes only those individual items for which a statistically significant difference was observed between groups; b—comparing intervention to control group participants, in constrained linear mixed models accounting for visit and arm by visit interaction.

## Data Availability

Data are unavailable for sharing due to privacy reasons – participants did not provide consent for data to be shared outside of the study.

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
