# Peer review of "A Technology-Driven, Healthcare-Based Intervention to Improve Family Beverage Choices: Results from a Pilot Randomized Trial in the United States"

_nutrients, 2023, doi:10.3390/nu15092141_

Round 1
Reviewer 1 Report
· The title should include the country where the study was conducted.
· Abstract: Please provide details about the age of participants, the place where the study was performed, the date of sampling took place, the type of analysis used and the significant p-value.
· Abstract: “These findings suggest that….beverage choice”. This statement should be clarified. Please elaborate more on the terms “feasible” and “multicomponent”.
· Please expand the keywords list. For example, “intervention”, “family”, “RCT”, “USA”…etc.
· Study design: “whose children are patients” Please clarify- patients with diabetes, cancer…etc.
· How feasible is it for you to bring about changes in knowledge, attitudes, and beliefs around family beverage choices? Need to explain how you are going to make this happen. How realistic/achievable are these outcomes? What evidence can you provide to the reader that you can feasibly achieve each of these?
· Table 1 should be moved to results.
· Table 3 is unclear. It seems that the authors used ANOVA/Fisher's analysis of variance to compare the mean change. The significant/non-significant P value should be clearly presented.
· I suggest that authors should add a table on the parental knowledge, attitude and belief outcomes.
· I suggest that authors should elaborate more on the sensitivity analyses section.
· Please follow the journal guideline for referencing.
Reviewer 2 Report
The authors are to be congratulated for attempting to mitigate a modern-day epidemic and it is a pity that the results are not more positive. The manuscript is well written and easy to read.
No aim is listed in the abstract and this should be fixed. In the discussion the aim is described as feasibility.
The inclusion criteria of 1-8 years old is very heterogeneous and the single biggest limitation of the study. A 1-year-old has no choice, and the study is effectively measuring parent compliance, Whereas, the 8 years olds that supply assent and may possibly make decisions without the knowledge of the parents or apply pressure to the parents to meet their desires. Unfortunately, this design feature can’t be changed.
I presume that 413 were approached in order to finish with a sample of 60. The response rate of 60/335 ~18% is very low and suggests that the results may be non -representative of the inferential population. Moreover, the non-random take-up might consist of those more likely to want to improve, so these results are probably optimistic.
The primary outcome in frequency was 2.4-1.3 =1.1 in the intervention and 2.6-1.5 = 1.1 in the controls so not difference at all. This is by definition also not clinically significant, nor is it due to the pilot nature of the study. The change in water intake = 19.7-16.6= 3.1 among the intervention and 16.7-15.5 = 1.2 among the controls, so the intervention seemed to have the effect of getting intervention participants to drink more water.
My take is that the improvement in controls over time is just due to the heightened awareness by being in trial.
As a minor point the authors should have a footnote in table 1 indicating that there is no residual confounding present.
Round 2
Reviewer 1 Report
No further comments.
Reviewer 2 Report
No more to add.